# Synchronous Pancreatic Ductal Adenocarcinoma in the Head and Tail, a Double Trouble: A Case Report and Literature Review

**DOI:** 10.3390/diagnostics12112709

**Published:** 2022-11-05

**Authors:** Daniel Paramythiotis, Georgia Fotiadou, Eleni Karlafti, Ioanna Abba Deka, Georgios Petrakis, Elisavet Psoma, Xanthippi Mavropoulou, Filippos Kyriakidis, Smaro Netta, Stylianos Apostolidis

**Affiliations:** 1First Propaedeutic Surgery Department, University General Hospital of Thessaloniki AHEPA, Aristotle University of Thessaloniki, 54634 Thessaloniki, Greece; 2Emergency Department, University General Hospital of Thessaloniki AHEPA, Aristotle University of Thessaloniki, 54634 Thessaloniki, Greece; 3First Propaedeutic Department of Internal Medicine, University General Hospital of Thessaloniki AHEPA, Aristotle University of Thessaloniki, 54634 Thessaloniki, Greece; 4Pathology Department, University General Hospital of Thessaloniki AHEPA, Medical School, Aristotle University of Thessaloniki, 54124 Thessaloniki, Greece; 5Radiology Department, University General Hospital of Thessaloniki AHEPA, Aristotle University of Thessaloniki, 54634 Thessaloniki, Greece; 6Second Chemotherapy Department, Theagenio Cancer Hospital of Thessaloniki, 54639 Thessaloniki, Greece

**Keywords:** synchronous primary, multifocal, PDAC, diabetes mellitus, pancreatectomy

## Abstract

Synchronous primary pancreatic ductal adenocarcinoma (PDAC) is very rare and can be formed either through multicentric carcinogenesis or intrapancreatic metastasis. We report the case of an 80-year-old man with a history of type 2 diabetes mellitus who presented with abdominal pain and weight loss. Laboratory tests showed elevated levels of blood glucose and CA 19-9, and Computed Tomography revealed two hypoenhancing lesions in the head and tail of the pancreas. Endoscopic ultrasound, which is the imaging method of choice for pancreatic cancer, was performed with a fine needle biopsy, and the cytological analysis diagnosed PDAC in both lesions. The patient underwent total pancreatectomy, and pathologic evaluation revealed synchronous primary PDAC with moderate to poor differentiation in the head and tail in the setting of IPMN (intraductal papillary mucinous neoplasia) and chronic pancreatitis. After his recovery from postoperative pulmonary embolism, the patient was discharged home with sufficient glycemic control. Multifocal PDAC occurs more often when precursor lesions, such as IPMN, pre-exist. The optimal treatment for multiple lesions spread all over the pancreas is total pancreatectomy. Diabetes mellitus is a serious complication of total pancreatectomy (new-onset or type 3c), but overall, long-term survival has been significantly improved.

## 1. Introduction

Pancreatic cancer, which predominantly presents as ductal adenocarcinoma, is one of the deadliest malignancies. According to the statistics, it is the tenth most common type of cancer and the fourth leading cause of cancer death in the United States [1,2]. In most cases, pancreatic cancer occurs as a single lesion [3], though it is possible to present as synchronous multifocal tumors. These tumors can either be metastatic or primary [4]. Synchronous primary pancreatic ductal adenocarcinoma (PDAC) is quite a rare entity [3,4,5] and is defined as the presence of two or more malignancies simultaneously in the pancreatic parenchyma, each of them developing independently from the others [6,7].

Studies suggest two possible ways of forming synchronous multifocal tumors: multicentric carcinogenesis and intrapancreatic metastasis [8]. The differentiation between them is not an easy task. In the first case, there are no common mutations between the lesions, that appear discontinuous from one another [3]. On the other hand, tumors resulting from intrapancreatic metastasis have some identical gene mutations and are associated with a worse prognosis [9,10].

We provide herein a case report of a synchronous primary pancreatic ductal adenocarcinoma detected in the head and tail of the pancreas, as well as a literature review regarding the diagnosis and treatment of pancreatic cancer.

## 2. Case Report

An 80-year-old man was admitted to our department for further investigation and treatment of two pancreatic lesions that were found in an abdominal computed tomography scan (CT) (Figure 1). The patient reported abdominal discomfort and an unexplained weight loss of 15 kg, during the last six months. His medical history includes an appendectomy, pilonidal cyst surgery, a healed gastric ulcer, coronary artery bypass surgery, coronary heart disease, hyperuricemia, hypothyroidism due to de Quervain thyroiditis, hypertension, type 2 diabetes mellitus, and dyslipidemia. He quit smoking thirty years ago, has no history of alcohol abuse, no allergies, and a family history of lung cancer. During the physical examination, there were no abnormal findings.

On his admission, laboratory tests including blood count, and liver and renal function indicators were within the normal limits, whereas blood glucose levels were found to be elevated (293 mg/dL). Regarding serum tumor markers, carbohydrate antigen 19-9 (CA 19-9) was outside the normal range (711.6 U/mL, normal values < 37 U/mL), carbohydrate antigen 72-4 (CA 72-4) was slightly elevated (11.86 U/mL, normal values < 7 U/mL), and carcinoembryonic antigen (CEA) was within the normal levels (3.45 ng/mL, normal values < 5 ng/mL).

A new abdominal contrast-enhanced computed tomography scan (CECT), which confirmed the findings of the first one, revealed two ill-defined hypoenhancing lesions with a maximum diameter of 2.5 cm in the head and tail of the pancreas, obstruction and dilatation of the main pancreatic duct, and abutment of the superior mesenteric vein <90° by the head tumor, as well as infiltration of the wall of the splenic vein by the tail tumor. Hepatic and renal cysts and an enlarged mesenteric lymph node were also depicted. The main pancreatic duct and its branch ducts appeared anomalously dilated with multiple stenoses (Figure 2). These findings were considered suspicious for double pancreatic adenocarcinoma in the setting of chronic pancreatitis or intraductal papillary mucinous neoplasia (IPMN).

Endoscopic ultrasound of the pancreas showed hypoechoic masses in the head and tail measuring 2.2 cm × 2.5 cm and 2.5 cm × 2.8 cm, respectively. We performed an endoscopic ultrasound-guided fine needle biopsy (EUS-FNB), and the cytological analysis revealed moderately differentiated PDAC in specimens from both the head and tail of the pancreas.

For further investigation and a better depiction, a contrast-enhanced magnetic resonance imaging/magnetic resonance cholangiopancreatography (MRI/MRCP) was performed, which showed an ill-defined hypointense lesion in the head of the pancreas with restricted diffusion. The main pancreatic duct (MPD) and its branch ducts appeared significantly dilated, mainly in the pancreatic head and tail (Figure 3 and Figure 4). In addition to this, MRI/MRCP demonstrated the presence of IPMN in the pancreatic parenchyma (Figure 5).

Due to the high malignant nature and the bifocal site of PDAC, as well as the infiltration of the splenic vein, the patient underwent total pancreatectomy with splenectomy, end-to-side choledochojejunostomy, and gastrojejunostomy.

Macroscopic examination of the two surgical specimens, the Whipple’s resection (including the pancreatic head) and the peripheral pancreatectomy (with the body and tail), revealed a neoplasm of diameter 3.5 cm in the pancreatic head and a second neoplasm of diameter 4.4 cm in the pancreatic tail.

Microscopically, both neoplasms were ductal adenocarcinomas, with moderate to poor differentiation and a substantial extent of clear cell morphology (clear cell subtype) (Figure 6) [11].

Immunohistochemically, the neoplastic cells were positive for CA19-9, CEA, CK7, and to a lesser degree, CA125 (Figure 7), while being negative for CK20.

Areas with low- and high-grade pancreatic intraepithelial neoplasia (PanIN) and findings of chronic pancreatitis with parenchymal architectural distortion, atrophy, fibrosis, and inflammation were also observed. Sections from the gallbladder, the spleen, and the overall surgical margins were free of neoplastic cells. One perisplenic lymph node was positive for metastasis, and the remaining 13 lymph nodes of both surgical specimens were free of metastases, leading to a pT2N0 stage for the first and a pT3N1 stage for the second adenocarcinoma, respectively.

The possibility of metastasis is considered in the differential diagnosis when multiple foci of tumors are encountered. Moreover, in this case, there was the unusual histological finding of some neoplastic cells exhibiting clear cell morphology. The presence of PanIN in the nearby parenchyma and the immunohistochemical findings of both tumors ruled out the possibility of metastasis to the pancreas of another tumor with clear cell morphology, namely a renal cell carcinoma, which would have been double negative for the CKs used and would not express the rest of the immunohistochemical markers.

On the sixth postoperative day, the patient developed hypoxemia and was diagnosed with femoral-popliteal deep venous thrombosis (DVT) and pulmonary embolism. He was treated with anticoagulants and moved out of the bed gradually. After the perioperative glucose management education, glycemic control was finally successful, and the patient was discharged on the sixteenth postoperative day. To minimize the risk of recurrence, adjuvant chemotherapy with gemcitabine was initiated. Five months after surgery, the patient was in good health, and no recurrence was found.

## 3. Discussion

The incidence of PDAC is constantly increasing [6], and the median age of diagnosis is 66 years [12]. According to the National Comprehensive Cancer Network (NCCN), the most important risk factors for developing a PDAC are smoking, obesity, alcohol abuse, diabetes mellitus, chronic pancreatitis, and pancreatic neoplastic cystic lesions, such as intraductal papillary mucinous neoplasms (IPMNs) and mucinous cystic neoplasia (MCN) [13,14]. Chronic pancreatitis and neoplastic cystic lesions, as well as pancreatic intraepithelial neoplasia (PanIN), are considered precursor lesions for pancreatic cancer [15,16], and they are often detected simultaneously with primary tumors [9]. Therefore, it is essential to depict the whole pancreas and search for a possible adenocarcinoma when a precursor lesion is detected [6,17].

PDAC is most often located in the head of the pancreas (71%), and rarely in the body (13%) or the tail (16%) [18]. Depending on the localization, the patient may experience different symptoms. Head tumors can cause jaundice due to the obstruction of the biliary system, whereas body or tail tumors present more often with abdominal or back pain [19]. In either case, PDAC can cause non-specific symptoms such as anorexia, weight loss, nausea, fatigue, or thrombophlebitis [20].

Much research has been conducted to determine the correlation between tumor localization and prognosis. On the one hand, head tumors may be associated with longer overall survival due to more specific symptoms and, therefore, earlier diagnosis, while tail tumors are often larger and more aggressive [21,22]. On the other hand, tail tumors are linked to a lower lymph node ratio (the number of positive lymph nodes to the number of examined lymph nodes) and higher disease-specific survival rates than head tumors of the same stage of cancer [18,23,24,25].

The prognosis of PDAC is also affected by the molecular characteristics of tumors. The most commonly mutated genes are KRAS, TP53, CDKN2A, and SMAD4. However, other factors, such as microenvironmental interactions, determine the biological behavior of tumors as well [26]. The molecular classification of the tumor could contribute to deciding on the best and most appropriate treatment for the patient [27].

Since the development of PDAC from an initial gene mutation can last up to ten years [16], it is of utmost importance to optimize diagnostic tools and, thus, ensure the early detection of the tumor. Among serum markers, CA 19-9 (carbohydrate antigen 19-9) is the only well-established tumor marker for PDAC. Although its diagnostic value is restricted due to low sensitivity and specificity, it can be used in symptomatic patients as a diagnostic tool when combined with imaging methods [28,29]. Multi-detector computed tomography (MDCT) is the standard imaging method, and the PDAC typically appears as an ill-defined hypoenhancing mass [30]. Magnetic resonance imaging (MRI) can be useful in detecting metastasis [31,32]. However, the diagnostic tool with the greatest sensitivity is endoscopic ultrasound (EUS), which also offers the possibility of receiving tissue samples through fine-needle aspiration (FNA) or fine-needle biopsy (FNB). EUS seems to be the ideal method for detecting small lesions (<2 cm) and determining lymph node involvement [30,33]. Nevertheless, performing EUS can result in bleeding or tumor seeding in the gastric walls [30,34]. Therefore, it is essential to establish new non-invasive diagnostic techniques with high specificity and sensitivity, such as immune-Positron Emission Tomography.

PDAC can appear simultaneously with primary malignancies in other organs, and the location of the other neoplasm can affect the prognosis [35,36]. It can also occur with other pancreatic lesions, more often in the context of a familial history of cancer. Multifocal tumors of the pancreas are highly suspicious of developing malignancy [37,38]. Hence, there is a need for differential diagnosis between PDAC and multifocal lesions, which usually consist of PanIN and IPMNs, but also between PDAC and chronic pancreatitis [16]. Mass-forming chronic pancreatitis can imitate both the symptoms and the imaging findings of pancreatic cancer. EUS-FNB can contribute to the correct diagnosis and prevent unnecessary surgical resection [20]. Another situation that must be distinguished from PDAC is autoimmune pancreatitis, which can rarely present with multifocal lesions [39,40].

In the patients diagnosed with pancreatic cancer, diabetes mellitus (DM) may either pre-exist years before (long-standing type 2 DM) or occur within the last 3 years before the cancer diagnosis (new-onset DM) [41]. New-onset DM, also known as type 3c DM, is induced by a disease or surgical resection of the pancreatic parenchyma [42,43]. Compared to long-standing DM, it is linked to lower overall survival rates [44], but blood glucose levels were significantly improved after pancreatectomy [45,46,47]. Moreover, new-onset DM acts protectively against chemotherapy, as these patients are more resistant to neutropenia [48]. To manage cancer-related DM, metformin was considered the treatment of choice, as it was found to suppress tumor development [49,50]. However, new studies have shown that insulin may have antineoplastic action as well [51].

The first step in treating PDAC is to determine the resectability of the tumor. According to a classification by the NCCN, PDAC may be characterised as resectable, borderline resectable, or unresectable [13,52]. The most common metastatic sites of primary PDAC are the liver, peritoneum, lung, extra-regional lymph nodes, and bones [53,54].

Resection is the treatment of choice for pancreatic cancer [55]. Depending on the location of the tumor, the patient might undergo a Whipple procedure (pancreaticoduodenectomy) for the head tumors, left or distal pancreatectomy for the body and tail tumors, or a total pancreatectomy for extended disease [56]. Thus, the occurrence of multifocal lesions spread all over the pancreatic parenchyma is an indication for total pancreatectomy [57,58,59]. Nevertheless, the decision on surgery is not always a simple task due to serious complications. As both the endocrine and exocrine portions of the pancreas is resected, the patients may suffer not only from postoperative DM, which requires insulin treatment, but also from exocrine pancreatic insufficiency with steatorrhea, weight loss, malabsorption, and dumping syndrome [57,58,60].

To prevent these complications, middle-preserving pancreatectomy could be an alternative solution under the condition that invasive lesions are detected only in one part of the pancreas, either the head or the tail [61]. Hence, preoperative staging of all lesions can be very useful for the clinical decision on surgery [62]. In addition, this parenchyma-sparing resection reduces the rates of postoperative hypoglycemia that total pancreatectomy usually causes [63,64,65].

Overall, total pancreatectomy is the surgical procedure of choice for synchronous primary PDAC in the head and tail [57,58,59], and the rates of postoperative mortality and long-term survival have been remarkably improved in recent years, especially when the surgery is performed in high-volume centers by experienced surgeons [59,66,67]. In our case, the patient, despite his age, was a good candidate for total pancreatectomy, as the disease was considered resectable and he was at low surgical risk.

An important prognostic factor is the lymph node ratio, which is strongly affected by the number of dissected and examined lymph nodes [68]. Song et al. [69] suggested that the minimum number of examined lymph nodes for a safe prediction of the prognosis should be six. On the other hand, a lymph node yield of 12 or more seems to be more acceptable and is indicated in [16].

Regarding chemotherapy, adjuvant therapy with FOLFIRINOX (5-FU, leucovorin, irinotecan, and oxaliplatin) is the optimal treatment for borderline resectable or non-resectable disease [13,70]. Gemcitabine and paclitaxel are considered alternatives [16]. For patients diagnosed with a resectable disease like our case, gemcitabine- or 5-FU-based adjuvant chemotherapy can prolong survival time by 2–5 months [70]. Neoadjuvant chemotherapy may offer a better outcome and improve resectability for the above-mentioned patients [71]. Future strategies aim to develop targeted therapies, such as KRAS inhibitors or TP53 reactivators, that will contribute to a more individualized treatment plan [72].

We gathered 18 cases of synchronous primary PDAC reported in the literature (Table 1) [3,7,8,9,17,60,73,74,75,76,77,78]. In most of the cases, there were two lesions in the pancreatic parenchyma [8,9,17,60,73,74,75,76,77,78], whereas more synchronous tumors were extremely rare [3,7]. Like our case, it is common that CA 19-9 levels are elevated [3,7,8,17,74,75], but unfortunately there is little evidence of glucose disorders and control [7,60,73,77]. In general, the correlation between PDAC and precursor lesions, such as PanIN, IPMNs, and pancreatitis, is confirmed through this literature review [3,8,9,74,75,76,78]. Unfortunately, PanIN is difficult to identify preoperatively with current diagnostic methods; thus its role in surveillance schemes is questionable, except in cases with partially resected pancreas.

## 4. Conclusions

We reported a rare case of a patient with synchronous primary PDAC in the head and tail who was treated with total pancreatectomy and whose postoperative blood glucose levels were successfully controlled with insulin. Overall, synchronous primary invasive tumors of the pancreas are rarely presented in the literature, as PDAC occurs more often with non-invasive lesions such as IPMNs. Future studies should focus on prevention and earlier diagnosis of this fatal disease, for example by discovering biomarkers for population screening.

## Figures and Tables

**Figure 1 diagnostics-12-02709-f001:**
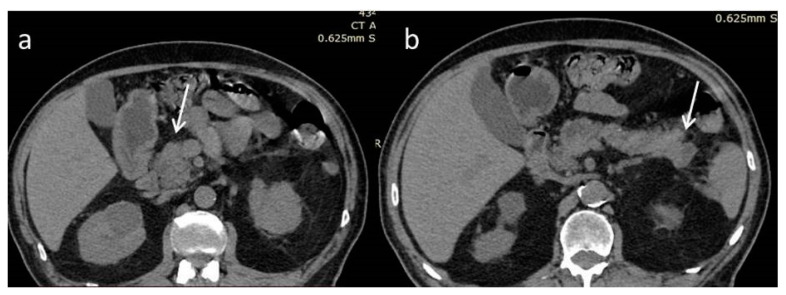
Non-enhanced computed tomography scan (NECT). Hypodense, poorly defined masses at the head (**a**) and tail (**b**) of the pancreas (white arrows).

**Figure 2 diagnostics-12-02709-f002:**
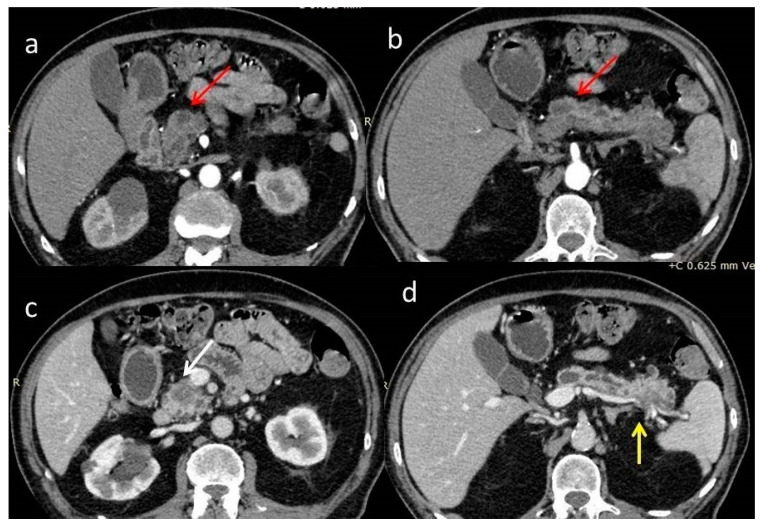
(**a**,**b**) arterial phase and (**c**,**d**) venous phase of a contrast-enhanced computed tomography scan (CECT). Poorly enhanced masses at the head and tail (red arrows) of the pancreas with surrounding fat stranding. The mass at the head of the pancreas occludes the pancreatic duct, causing peripheral dilatation (white arrow). Limited contact with the posterior surface of the superior mesenteric vein <90°. The mass of the tail invades the splenic vein (yellow arrow). Abnormal dilatation of the main pancreatic duct with strictures is demonstrated with multiple collateral cystic lesions.

**Figure 3 diagnostics-12-02709-f003:**
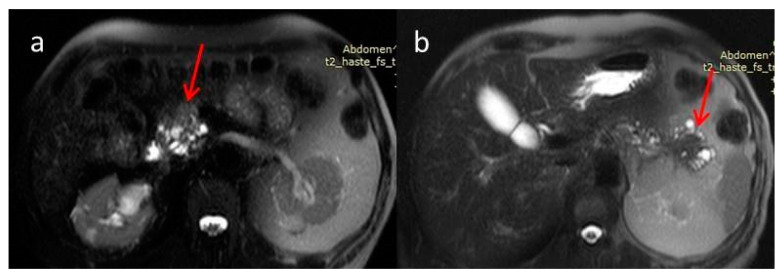
Magnetic resonance imaging T2-weighted image (MRI T2WI). The masses of the head (**a**) and tail (**b**) appear with low signal intensity (red arrows). Dilatation of the main duct is demonstrated with multiple dilated side branches. The overlying pancreatic parenchyma is thinned.

**Figure 4 diagnostics-12-02709-f004:**
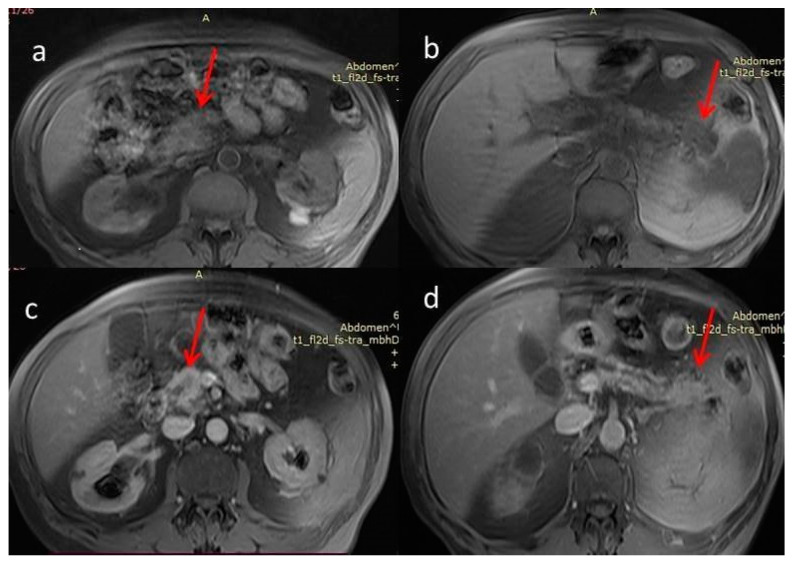
(**a**,**b**) Magnetic resonance imaging fat-suppressed T1 weighted image (MRI T1WI FS). Hypointense masses. (**c**,**d**) Magnetic resonance imaging with fat-suppressed post-Gadolinium T1 (MRI T1GD FS). Hypoenhancement of the masses compared to the normal pancreas (red arrows).

**Figure 5 diagnostics-12-02709-f005:**
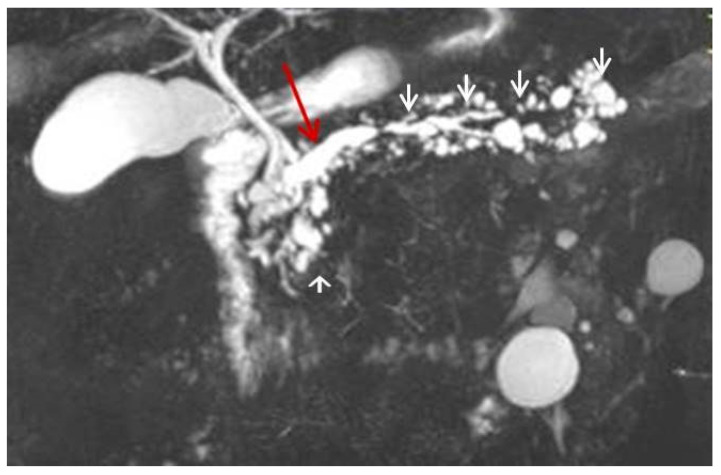
Magnetic resonance cholangiopancreatography (MRCP) exhibits the mixed type of IPMN, with an abnormal dilation of the Wirsung duct (red arrow) and multiple asymmetrically dilated side branches (white arrowheads).

**Figure 6 diagnostics-12-02709-f006:**
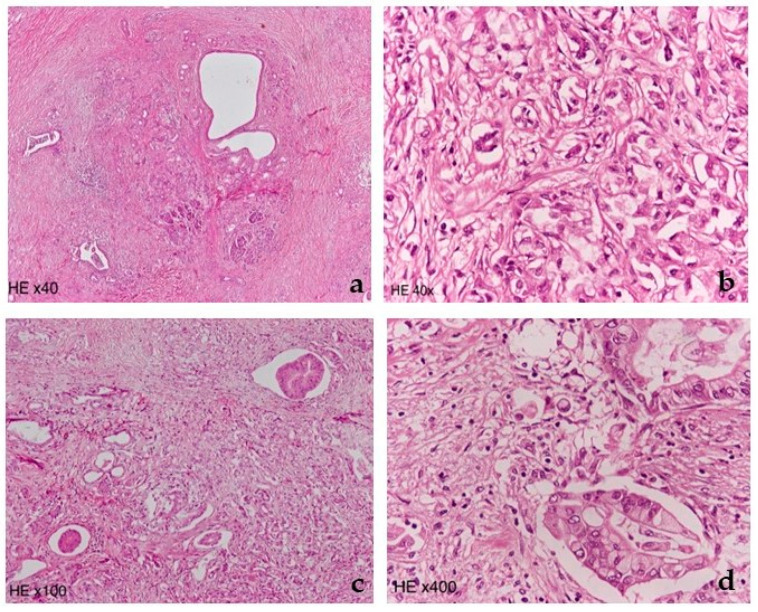
(**a**,**b**) Pancreatic head tumor; (**c**,**d**) Pancreatic tail tumor. Hematoxylin and eosin stains, (40×–400× magnifications), show the two moderately to poorly differentiated adenocarcinomas of the pancreatic head and tail at different magnifications. The atypical neoplastic glandular structures surrounded by desmoplastic stroma can be appreciated in the (**a**) head and (**c**) tail in low magnification. The neoplastic cells with irregular atypical nuclei, prominent nucleoli, and abundant eosinophilic and clear cytoplasm can be appreciated in (**b**) head and (**d**) tail in high magnification.

**Figure 7 diagnostics-12-02709-f007:**
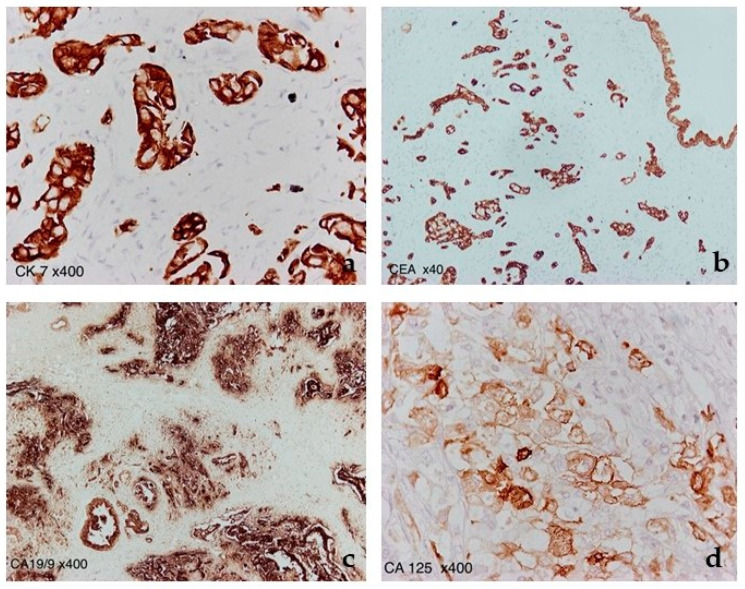
Immunostaining for (**a**) CK7 (400× magnification), (**b**) CEA (40× magnification), (**c**) CA19-9 (400× magnification), and (**d**) CA125 (400× magnification), highlighting the neoplastic glands of both adenocarcinomas.

**Table 1 diagnostics-12-02709-t001:** Summary of the cases of synchronous primary pancreatic ductal adenocarcinoma reported in the literature.

	Author/Year	Sex/Age	Location of PDAC	Number of Lesions	Initial Blood Glucose Levels/Diabetes Mellitus	Clinical Presentation
1.	Siassi et al./1999 [73]	F/62	body, tail	2	normal/new-onset due to pancreatic resection	jaundice
2.	Izumi et al./2009 [3]	F/75	head, head, body, body	4	NA	back pains
3.	Koizumi et al./2009 [74]	M/52	head, tail	2	NA	jaundice
4.	Fujimori et al./2010 [7]	M/77	head, body, tail	3	166 mg/dL fasting glucose/type 2	(-)
5.	Mori et al./2010 [75]	M/57	head, tail	2	NA/type 2	NA
6.	Kyokane et al./2011 [76]	F/71	body, tail	2	NA	back pains
7.	Goong et al./2015 [17]	F/61	head, tail	2	NA	abdominal discomfort, jaundice
8.	McGregor et al./2018 [77]	M/72	head, tail	2	normal/new-onset due to pancreatic resection	NA
9.	Sugiura et al./2019 [78]	F/69	body, tail	2	NA	NA
10.	Fujita et al./2020 [9]	NA	body, tail	2	NA	NA
11.	Fujita et al./2020 [9]	NA	body, tail	2	NA	NA
12.	Fujita et al./2020 [9]	NA	body, tail	2	NA	NA
13.	Fujita et al./2020 [9]	NA	body, body	2	NA	NA
14.	Fujita et al./2020 [9]	NA	body, tail	2	NA	NA
15.	Fujita et al./2020 [9]	NA	body, tail	2	NA	NA
16.	Fujita et al./2020 [9]	NA	body, tail	2	NA	NA
17.	Nitta et al./2020 [60]	F/77	head, tail	2	new-onset due to resection	NA
18.	Ohike et al./2020 [8]	F/70	body, tail	2	NA	NA
	**Tumor Markers in Serum**	**Endoscopic Findings**		**Biopsy**	**Surgery**	**Precursor Lesions**
1.	CEA, CA 19-9 normal	MPD splitting in the tail, MPD obstruction near the ampulla of Vater		negative for malignancy (CT-guided FNAB)	distal pancreatectomy, pylorus-preserving partial PD	NA
2.	CA 19-9, SPAN-1 elevated	MPD stenosis in the body and tail		NA	pylorus-preserving subtotal PD	PanIN
3.	CEA, CA 19-9 elevated	hypoechoic masses		NA	TP	IPMN
4.	CA 19-9, sIL-2R elevated	MPD dilatation and hypoechoic lesions		NA	TP	(-)
5.	CA 19-9 elevated	dilatation of the branch duct in the tail, stenotic lesion in the MPD in the body		ADC (ERCP cytology)	TP	IPMN
6.	CEA, CA 19-9 normal	NA		NA	distal pancreatectomy	PanIN
7.	CA 19-9 elevated	hypoechoic masses		PDAC (EUS-FNB)	(-)	(-)
8.	NA	NA		PDAC (EUS-FNB)	TP	NA
9.	NA	hypoechoic masses and atrophic pancreatic parenchyma between them		ADC (EUS-FNA)	(-)	pancreatitis
10.	NA	NA		NA	pancreatectomy	PanIN
11.	NA	NA		NA	pancreatectomy	PanIN
12.	NA	NA		NA	pancreatectomy	PanIN
13.	NA	NA		NA	pancreatectomy	(-)
14.	NA	NA		NA	pancreatectomy	(-)
15.	NA	NA		NA	pancreatectomy	PanIN
16.	NA	NA		NA	pancreatectomy	(-)
17.	normal	NA		ADC (EUS-FNA)	middle segment-preserving pancreatectomy	NA
18.	CA 19-9 elevated	NA		ADC (EUS-FNA)	distal pancreatectomy	PanIN
	**Maximum Diameter of Tumor (mm)**	**Metastatic Lymph Nodes**	**Stage**	**Glycemic Control**	**Chemotherapy**	**Recurrence/Outcome (Months)**
1.	10 (body), 50 (tail)	(-)	I	insulin, glibenclamide	NA	(-)/Survival (12)
2.	25 (head), 20 (head), 10 (body), 10 (body)	NA	NA	NA	S-1 (adjuvant)	(+)/Survival (6)
3.	25 (head), 35 (tail)	(+)	IVb, III	NA	NA	(-)/Survival (11)
4.	20 (head), 35 (body), 15 (tail)	1	IIB	insulin	gemcitabine (adjuvant)	(-)/Survival (12)
5.	12 (head), 3 (tail)	1	NA	NA	gemcitabine (adjuvant)	(-)/Survival (6)
6.	35 (body), 20 (tail)	(+)	IVa	NA	gemcitabine, TS-1 (adjuvant)	NA/Survival (18)
7.	49 (head), 24 (tail)	(+)	IIB	NA	chemoradiotherapy	NA
8.	13 (head), 14 (tail)	(-)	NA	insulin pump	FOLFIRINOX (neoadjuvant)	(-)/Survival (39)
9.	35 (body), 23 (tail)	NA	IV	NA	(palliative)	NA
10.	16 (body), 29 (tail)	NA	IIB	NA	NA	(-)/Survival (53)
11.	20 (body), 30 (tail)	NA	IIB	NA	NA	(+)/Survival (48)
12.	8 (body), 24 (tail)	NA	IIB	NA	NA	(-)/Death (50)
13.	12 (body), 1 (body)	NA	IIB	NA	NA	(+)/Death (44)
14.	35 (body), 1 (tail)	NA	III	NA	NA	(+)/Death (27)
15.	32 (body), 1 (tail)	NA	III	NA	NA	(+)/Death (35)
16.	7 (body), 30 (tail)	NA	III	NA	NA	(+)/Death (15)
17.	18 (head), 32 (tail)	(+)	IIB/ IB	dipeptidyl peptidase-4 inhibitor	S-1 (adjuvant)	(-)/Survival (9)
18.	19 (body), 45 (tail)	(-)	IIA/ IA	NA	NA	(+)/Death (65)

Abbreviations: F: female; M: male; PDAC: pancreatic ductal adenocarcinoma; NA: not answered; (-)/(+): absent/present; CEA: Carcinoembryonic antigen; CA19-9: Carbohydrate antigen 19-9; sIL-2R: soluble IL-2 receptor; MPD: main pancreatic duct; CT: Computed tomography; FNAB: fine needle aspiration biopsy; ADC: adenocarcinoma; ERCP: Endoscopic retrograde cholangiopancreatography; EUS-FNA/FNB: Endoscopic ultrasound fine needle aspiration/biopsy; PD: pancreaticoduodenectomy; TP: total pancreatectomy; PanIN: pancreatic intraepithelial neoplasia; IPMN: intraductal papillary mucinous neoplasm; S-1/TS-1: tegafur, gimeracil, and oteracil potassium; FOLFIRINOX: 5-FU, leucovorin, irinotecan, and oxaliplatin.

## Data Availability

Not applicable.

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
