# Peer review of "Synchronous Pancreatic Ductal Adenocarcinoma in the Head and Tail, a Double Trouble: A Case Report and Literature Review"

_diagnostics, 2022, doi:10.3390/diagnostics12112709_

Round 1

Reviewer 1 Report

l  In the “Case Report” section, the image that the authors described in line 75 that should be Figure 2, but the brackets indicated “Figure 1”.

l  In the “Case Report” section, the authors said “Endoscopic ultrasound of the pancreas showed hypoechoic masses in the head and tail measuring 2.2 cm x 2.5 cm and 2.5 x 2.8 cm, respectively” in line 94. Please keep the unit consistent between “2.2 cm x 2.5 cm” and “2.5 x 2.8 cm”.

l  In the “Case Report” section, the authors said “a peripheral pancreatectomy (with the body and tail) revealed a neoplasm of diameter 3.5 cm in the pancreatic head and a second neoplasm of diameter 4.4 cm in the pancreatic tail” in line 121, the word “a” should be revised as “the”.

l  In the “Discussion” section, the authors said “New-onset DM Compared to long-standing DM, ..., but blood glucose levels were significantly improved after pancreatectomy [45–47]” in line 215. Are you sure? Check the accuracy of reference 45-47. Your descriptions in the main text are not consistent with the original descriptions in these references.

l  In the “Discussion” section, the authors said “Moreover, new-onset DM acts protectively against chemotherapy, as these patients are more resistant to neutropenia” in line 217. The original reference explored diabetes, rather than new-onset diabetes. Check the accuracy of your reference.

l  In the “Discussion” section, the authors said “Therefore, multifocal lesions spread all over the pancreatic parenchyma are an indication of total pancreatectomy” in line 229. This sentence is wrong in grammar.

l  Was it appropriate to operate on an 80-year-old patient with pancreatic cancer? Was this patient suitable for chemotherapy after surgery?

l  What was the survival time? Was his life expectancy extended by surgery?

l  The authors should pay more attention on the features and outcomes of this patient in the Discussion section.

Author Response

Response to Reviewer 1 Comments

First of all, we would like to thank you for the detailed review and the valuable comments, which are really helpful in improving our manuscript. We address our responses for all of your referees below.

Point 1: In the “Case Report” section, the image that the authors described in line 75 that should be Figure 2, but the brackets indicated “Figure 1”.

Response 1: We apologize for the mistake. We have already corrected it.

Point 2: In the “Case Report” section, the authors said “Endoscopic ultrasound of the pancreas showed hypoechoic masses in the head and tail measuring 2.2 cm x 2.5 cm and 2.5 x 2.8 cm, respectively” in line 94. Please keep the unit consistent between “2.2 cm x 2.5 cm” and “2.5 x 2.8 cm”.

Response 2: Thank you for noticing. We have already corrected it.

Point 3: In the “Case Report” section, the authors said “a peripheral pancreatectomy (with the body and tail) revealed a neoplasm of diameter 3.5 cm in the pancreatic head and a second neoplasm of diameter 4.4 cm in the pancreatic tail” in line 121, the word “a” should be revised as “the”.

Response 3: Thank you for the correction. It is now fixed.

Point 4: In the “Discussion” section, the authors said “New-onset DM Compared to long-standing DM, ..., but blood glucose levels were significantly improved after pancreatectomy [45–47]” in line 215. Are you sure? Check the accuracy of reference 45-47. Your descriptions in the main text are not consistent with the original descriptions in these references.

Response 4: Shingyoji et al. (reference 45 in manuscript) reported that the patients with new-onset diabetes mellitus due to pancreatic cancer, that presents before the surgery, “were found to have high probability of diabetes resolution after pancreaticoduodenectomy”. References 46 and 47 were replaced with more relevant ones. Roy et al. (reference 46 in manuscript) reported “a significant improvement in the glycaemic control (75%) or resolution of NOD (20%-65%) after pancreatic surgery.”, and Andersen et al. (reference 47 in manuscript) observed that “new-onset diabetes associated with PDAC may resolve following tumor resection, as long as there are sufficient islets left in the residual pancreatic tissue, whereas all of the patients with long-standing diabetes had a persistence of diabetes after pancreatic resection”.

Point 5: In the “Discussion” section, the authors said “Moreover, new-onset DM acts protectively against chemotherapy, as these patients are more resistant to neutropenia” in line 217. The original reference explored diabetes, rather than new-onset diabetes. Check the accuracy of your reference.

Response 5: The original reference compares the characteristics of patients without diabetes mellitus with those diagnosed with diabetes mellitus (either long-standing type 2 or new-onset DM). This study claims that patients diagnosed with DM are generally more resistant to neutropenia. However, as you can see in Table 1 of this reference, neutropenia as an adverse effect of adjuvant chemotherapy is less likely to occur in patients with new-onset DM than in those with long-standing DM (38.9% against 58.6%).

Point 6: In the “Discussion” section, the authors said “Therefore, multifocal lesions spread all over the pancreatic parenchyma are an indication of total pancreatectomy” in line 229. This sentence is wrong in grammar.

Response 6: We are sorry for the confusion. We have already corrected it.

Point 7: Was it appropriate to operate on an 80-year-old patient with pancreatic cancer? Was this patient suitable for chemotherapy after surgery?

Response 7: This patient was a good candidate for surgery because first of all the disease was resectable, and at the time of diagnosis no metastases were found. Secondly, the resection was suitable, because the patient was at low surgical risk, in very good physical condition, and without many comorbidities. Regarding chemotherapy, according to Adamska et al. (reference 70 in manuscript) “Surgery followed by adjuvant therapy has been shown to provide slight, but significant survival benefit for non-metastatic patients in several phase III studies. Thus far, gemcitabine and 5-FU-based postoperative chemoradiation has been considered as standard of care, improving the median OS time for 2–5 months”. Therefore, the patient was also suitable for adjuvant chemotherapy.

Point 8: What was the survival time? Was his life expectancy extended by surgery?

Response 8: At five months after surgery, the patient was in good health, and no recurrence was found. According to Johnston et al. [1], total pancreatectomy improved significantly the survival rates in patients with tumor size 2–5 cm, less than 3 positive lymph nodes, and negative surgical margins. Patients with age >70 years were less likely to benefit from the resection, but if other criteria, such as those previously mentioned, are met, the survival is improved.

Reference

  1. Johnston, W.C.; Hoen, H.M.; Cassera, M.A.; Newell, P.H.; Hammill, C.W.; Hansen, P.D.; Wolf, R.F. Total pancreatectomy for pancreatic ductal adenocarcinoma: review of the National Cancer Data Base. HPB: the official journal of the International Hepato Pancreato Biliary Association 2016, 18, 21–28, doi:10.1016/j.hpb.2015.07.009.

Point 9: The authors should pay more attention on the features and outcomes of this patient in the Discussion section.

Response 9: Some changes were made in the manuscript, and it is now more focused on our patient.

Reviewer 2 Report

The authors presented a rare case of synchronous primary pancreatic ductal adenocarcinoma (PDAC) and did a literature review relevant to it. In my opinion,  a literature review comparing the clinical information between synchronous and non-synchronous PDCA will be more informative than a summary of the clinical data of synchronous PDCA.

Author Response

Response to Reviewer 2 Comments

Point: The authors presented a rare case of synchronous primary pancreatic ductal adenocarcinoma (PDAC) and did a literature review relevant to it. In my opinion, a literature review comparing the clinical information between synchronous and non-synchronous PDCA will be more informative than a summary of the clinical data of synchronous PDCA.

Response: First, we would like to thank you for the careful review and thoughtful comment.

Since we didn’t find such a review previously written, we attempted to gather all the cases of synchronous primary pancreatic ductal adenocarcinoma (PDAC) that are reported in the literature. However, we agree that a literature review comparing the clinical information between synchronous and non-synchronous PDAC would be more interesting and informative. This time, unfortunately, we have had not plenty of time to write such an extensive review, but we could definitely submit later a paper on this topic.

Reviewer 3 Report

Case report is well presented.

Discussion is detailed and relevant.

Literature review of previous reported cases is detailed.

References are mostly recent and up to date.

Author Response

Response to Reviewer 3 Comments

Comment: Case report is well presented. Discussion is detailed and relevant. Literature review of previous reported cases is detailed. References are mostly recent and up to date.

Response: We would like to thank you for your review. We really appreciate the positive feedback.

Reviewer 4 Report

General: This is a report of a case involving synchronous pancreatic adenocarcinoma lesions – describing the clinical findings and summary of the analogously reported cases. Identification of multiple lesions within pancreas raises the question of primary synchronous versus metastatic lesions. The difficulty can be encountered in the distinguishing the two groups and suggested in the literature looking at the gene mutations.

Major issues: 

1.     No clear demonstration of “research findings and updated reviews on topics related to advanced diagnostic methods in gastrointestinal disease” as aimed in this Special Issue. Standard clinical methodologies for detection and diagnosis of pancreatic cancer were applied in the case. As mentioned in the Introduction section, genetic mutations may be elucidated to advance the methods or criteria to distinguish the metastatic and primary pancreatic lesions, which are not described for the current case. The distinguishing features by IHC markers are also not described in detail as to differentiating from the renal cell cancer.

2.     As IPMN was seen in this case, 2 cases in the reviewed literature were noted – though many were ‘NA’ or ‘-‘. A question of potentially increased occurrences of synchronous (or metachronous) pancreatic cancer maybe raised in IPMNs for the surveillance purposes (as many of these lesions are followed). PanINs are difficult to identify preoperatively with the current technology, so it is questionable as to its role in surveillance schemes except in cases noted previously with  partially resected pancreas.

Minor issues: None.

Author Response

Response to Reviewer 4 Comments

Firstly, we would like to thank you for the insightful review and the thoughtful comments, which are really helpful in improving our manuscript. We address our responses for all of your referees below.

General comment: This is a report of a case involving synchronous pancreatic adenocarcinoma lesions – describing the clinical findings and summary of the analogously reported cases. Identification of multiple lesions within pancreas raises the question of primary synchronous versus metastatic lesions. The difficulty can be encountered in the distinguishing the two groups and suggested in the literature looking at the gene mutations.

Response: As already stated in the manuscript, the possibility of metastasis is considered in the differential diagnosis when multiple foci of tumors are encountered. In our case, the presence of PanIN in the nearby parenchyma and the immunohistochemical findings of both tumors ruled out the possibility of metastasis to the pancreas of another tumor. The next step is to differentiate intrapancreatic metastasis from multicentric carcinogenesis. In this process, the identification of gene mutations in each lesion would be the only way to be sure of the diagnosis.

Point 1: No clear demonstration of “research findings and updated reviews on topics related to advanced diagnostic methods in gastrointestinal disease” as aimed in this Special Issue. Standard clinical methodologies for detection and diagnosis of pancreatic cancer were applied in the case. As mentioned in the Introduction section, genetic mutations may be elucidated to advance the methods or criteria to distinguish the metastatic and primary pancreatic lesions, which are not described for the current case. The distinguishing features by IHC markers are also not described in detail as to differentiating from the renal cell cancer.

Response 1: Standard methods for detection and diagnosis are basically used. Future studies should focus more on non-invasive techniques, such as immuno-Positron Emission Tomography. In our case, we agree that it would be necessary to examine if these genetic mutations were present on the tumor tissue, in order to distinguish a metachronous from a synchronous lesion. However, this is not common practice in our institution due to the cost of this diagnostic method in a low-income country, like Greece. Regarding differentiation from renal cell cancer (RCC), the IHC profile already stated in the manuscript is contradictory to renal cell cancer origin. RCCs are usually, if not always, characteristically double negative for CK7 and CK20. Moreover, the rest of the positive stains are also not encountered in RCCs.

Point 2: As IPMN was seen in this case, 2 cases in the reviewed literature were noted – though many were ‘NA’ or ‘-‘. A question of potentially increased occurrences of synchronous (or metachronous) pancreatic cancer maybe raised in IPMNs for the surveillance purposes (as many of these lesions are followed). PanINs are difficult to identify preoperatively with the current technology, so it is questionable as to its role in surveillance schemes except in cases noted previously with partially resected pancreas.

Response 2: Thank you for the critical comment. We have added the information to the manuscript. PanIN is indeed mainly diagnosed after surgery, and its role in surveillance schemes is restricted. However, it is mentioned because it is considered a risk factor for developing pancreatic cancer, like IPMNs. Regarding IPMNs, they usually preexist and require surveillance, even after their excision, on the one hand, due to their malignant potential, and on the other hand, due to a general “instability” of the pancreatic parenchyma, which could lead to the development of carcinoma in other pancreatic sites, either synchronous or metachronous [1].

Reference

  1. European Study Group on Cystic Tumours of the Pancreas. European evidence-based guidelines on pancreatic cystic neoplasms. Gut 2018, 67, 789–804, doi:10.1136/gutjnl-2018-316027.

Round 2

Reviewer 4 Report

This is a well-described clinical case report of an uncommon synchronous cancer lesions within pancreas. However, this does not seem to fit within the Aim for this Special Issue - “research findings and updated reviews on topics related to advanced diagnostic methods in gastrointestinal disease”. There appears to be no clear focus (including any detailed review on a topic) on any advanced diagnostic methods.